# Metabolic and Biochemical Responses of Juvenile *Babylonia areolata* to Hypoxia Stress

**DOI:** 10.3390/biology14060727

**Published:** 2025-06-19

**Authors:** Baojun Tang, Xiaoyao Ren, Zhiguo Dong, Hanfeng Zheng, Yujia Liu, Tao Wei

**Affiliations:** 1East China Sea Fisheries Research Institute, Chinese Academy of Fishery Sciences, Shanghai 200090, China; bjtang@yeah.net (B.T.); xyren1999@163.com (X.R.); liuyj2000@126.com (Y.L.); wt18020479686@163.com (T.W.); 2School of Marine Science and Fisheries, Jiangsu Ocean University, Lianyungang 222005, China; dzg7712@163.com

**Keywords:** hypoxia, *Babylonia areolata*, metabolic rates, enzyme activities, gene expression

## Abstract

The snail *Babylonia areolata* has important economic and ecological values, yet it frequently encounters varying degrees of hypoxic conditions during farming and transportation. These low-oxygen environments are influenced by factors such as high farming density, degradation of substrate quality, and transport without water. Currently, there is limited information regarding the metabolic and immune responses of *B. areolata* under hypoxic stress. Based on previous acute stress experiments, we investigated the effects of prolonged hypoxia on mortality, metabolic rates, enzyme activities related to metabolism and immunity, and gene expression in *B. areolata*. Our study demonstrated that *B. areolata* is sensitive to hypoxic stress, with mortality observed after 2–3 days of exposure to dissolved oxygen concentrations of 2 and 0.5 mg O_2_/L. Concurrently, both metabolic activity and biochemical functions were found to be suppressed. Furthermore, we discovered that *B. areolata* may respond to hypoxic stress through an anaerobic energy metabolic pathway. This study underscores the critical importance of maintaining appropriate levels of dissolved oxygen in aquatic environments and ensuring suitable transportation durations for this species.

## 1. Introduction

Dissolved oxygen (DO) is a critical environmental factor for most aquatic organisms. However, since the mid-20th century, the oxygen content in the ocean has decreased by approximately 2% [1]. The prevalence of hypoxic (<2 mg/L) and anoxic conditions in coastal and estuarine regions has risen since the mid-1900s. The formation of dead zones resulting from dissolved oxygen depletion in coastal waters poses significant consequences for the functioning of marine ecosystems [2]. Marine shellfish primarily inhabit estuaries and coastal waters, where they are subjected to periodic fluctuations in dissolved oxygen concentrations due to natural factors. Additionally, they face low-oxygen environmental stresses resulting from global climate change and anthropogenic activities throughout their life cycle [3]. Many marine shellfish serve as significant aquaculture species. During the processes of farming and transportation, these organisms frequently encounter low-oxygen environments caused by high farming densities, degradation of substrate conditions, and transport without water.

In order to enhance the chances of survival in hypoxic or anoxic environments, marine organisms utilize a variety of coping strategies, among which metabolic regulation is a crucial mechanism for adapting to low-oxygen conditions [4]. The response pattern of respiratory metabolism under hypoxic conditions reflects the tolerance and adaptability of shellfish to such environments. A number of studies have investigated the effects of hypoxia stress on the respiratory metabolism of various marine shellfish, including the scallop *Pecten maximus* [4], the mussel *Mytilus edulis* [5,6], the oyster *Crassostrea virginica* [3], the thick-shelled mussel *M. coruscus* [7], and the snail *Busycon contrarium* [8]. The studies have demonstrated that marine shellfish can delay the depletion of stored energy reserves by adjusting their metabolic patterns and reducing metabolic rates, thereby aiding in the maintenance of energy homeostasis and survival under hypoxic conditions [4,9,10]. Energy metabolism requires the involvement of cellular metabolic enzymes. Research has indicated that shellfish can respond to hypoxic stress by modulating the expression of enzymes associated with glycolysis and anaerobic metabolism, such as lactate dehydrogenase (LDH) and pyruvate kinase (PK) [10,11,12].

Many studies have reported increased superoxide anion and H_2_O_2_ production during hypoxia through a mechanism involving respiratory complex III [13]. Moreover, the stabilization and accumulation of HIF-1, which can induce the expression of genes related to energy production, such as erythropoiesis and glycolysis [14,15], also need ROS generated by mitochondrial complex III [16]. Hypoxia-induced ROS excess can cause cellular oxidation and cellular damage. Superoxide dismutase (SOD) plays an important role in catalyzing the decomposition of excessive superoxide. Acid phosphatase (ACP) and (AKP) are important multifunctional enzymes involved in metabolism and innate immunity in marine organisms [17,18]. Many studies have shown that ACP and AKP play an important role in the response of marine organisms to hypoxic stress. The level of ACP and AKP activities is closely related to the intensity of stress [15,17,19].

*Babylonia areolata* (Link, 1807) is mainly distributed in southeastern Asia, including Thailand, Vietnam, Malaysia, and the coastal zone of southern China. Because of its fast growth rate and rich unsaturated fatty acids content, it has become one of the major aquaculture shellfish species in the coastal areas of southern China, with an annual production value of 450 million—550 million US dollars [20]. The farming of *B. areolata* requires the maintenance of a high DO level (>6 mg/L) [21]. However, under high-density farming conditions, dissolved oxygen in water is easily depleted, resulting in hypoxic stress. Combined with environmental changes such as elevated temperatures, dissolved oxygen (DO) levels can decrease by more than 90%, leading to significant mortality rates. Furthermore, *B. areolata* is likely to encounter hypoxic conditions during long-distance transportation. It has been shown that 2 mg O_2_/L is an important threshold to cause changes in metabolism and feeding, and even mortality, in marine shellfish [22,23]. This was also confirmed in our previous acute hypoxia study [24], in which dissolved oxygen levels were reduced to 6, 4, 2, 1, and 0.5 mg O_2_/L within 30 min, and the oxygen consumption rate at 2 mg O_2_/L was not significantly different from those observed at both 4 and 6 mg O_2_/L. However, there was a significant decrease in the oxygen consumption rate when dissolved oxygen levels dropped to 1 mg/L, even though no mortality was observed in the acute experiments. Prolonged anaerobic metabolism is bound to cause damage to *B. areolata*, and it is worth investigating how the snails cope with this situation. There is a lack of comprehensive studies examining the effects of hypoxic stress on the respiratory metabolism and biochemical function of *B. areolata* [25]. In the present study, we analyzed the metabolic rates, associated enzyme activities, and gene expression levels in juvenile *B. areolata* under two different hypoxic conditions. This study offers valuable insights into the metabolic and immune responses to hypoxia in marine molluscs.

## 2. Materials and Methods

### 2.1. Experimental Snails

The snails were sourced from a *B. areolata* farm located in Qionghai City, Hainan Province, China, and were transported to the laboratory using a low-temperature incubator set at 4–10 °C for acclimation in a 400 L tank. The bottom of the tank was covered with approximately 7 cm of fine sand. During the acclimation period, the snails were fed trash fish at about 30% of their total body mass daily. To prevent deterioration of water quality, 50% of the water was replaced, and food residues were removed within one hour after feeding. The water temperature was maintained at 28 ± 1 °C, with salinity levels at 28‰ and pH measured at 8.1. Seawater aeration was implemented to ensure that dissolved oxygen (DO) levels remained above or equal to 7.5 mg/L. After an acclimation period of thirty days, snails with similar sizes (shell height: 19.89 ± 1.36 mm; wet weight: 1.58 ± 0.37 g) were selected for the experiment.

### 2.2. Hypoxic Stress and Sample Collection

The experiment was conducted using nine plastic drums (diameter: 39 cm, height: 55 cm), with seawater filtered through a sand filtration system. A total of 540 snails were evenly distributed into three groups, each consisting of three replicates. The dissolved oxygen (DO) level was gradually adjusted to 2.0 mg/L within half an hour and subsequently to 0.5 mg/L within one hour by means of nitrogen bubbling. The fluctuation range of DO was maintained within ±0.1 mg/L. For the control group, the DO level was sustained at 7.5 mg/L throughout the experiment. The time at which the DO level reached the predetermined value was recorded as 0 h, and the entire duration of the experiment spanned 144 h. To mitigate any deterioration in water quality, no diet was fed during the experiment, and seawater changes were performed every 72 h; additionally, replaced seawater was pre-adjusted to match the experimental DO levels prior to introduction into the drums. Dead individuals were examined and removed every 24 h. The dead state was defined as the absence of response to gentle pulling of the operculum with forceps, lack of contraction in the respiratory tube and ventral pedicle upon touch, or a distinct odor. Every 24 h, four snails were collected from each replicate drum and placed into respiration chambers for metabolic rate determination. The individuals used for respiration measurements were placed in a mesh bag and returned to the same drum. So, the same set of snails was measured each time. Simultaneously, two snails were taken from each replicate, and whole tissue samples were obtained and stored in a refrigerator at −80 °C for subsequent enzyme activity and gene expression analyses. In calculating the cumulative mortality rate, these snails sampled were included for ease of calculation, assuming that they would live until the end of the experiment.

### 2.3. Metabolic Rates Measurement

Oxygen consumption rate was measured in a glass respiration chamber with a volume of 0.56 L. The DO concentration was assessed using a Leici JPBJ-609L oxygen meter equipped with a DO-968-HC oxygen electrode (Inesa Scientific Instrument Co., Ltd., Shanghai, China). For each DO level, three replicates and one blank control were established. Juvenile snails were placed in a mesh bag and suspended in the respiration chamber to ensure their activities remained unaffected. A magnetic rotor was employed to achieve thorough mixing of the seawater within the chamber. The respiration chambers were sealed with rubber stoppers after being connected to the oxygen meter, allowing for continuous monitoring of DO concentrations over time. Measurements lasted for 1 h, during which readings were recorded at 5 min intervals, resulting in a total of 12 recordings per measurement session. Each measurement was conducted in triplicate, while an additional chamber without snails served as the blank control. At the conclusion of the experiment, water samples from the respiration chambers were collected for ammonia concentration analysis, which was performed using the phenol-hypochlorite method [26]. Additionally, the wet mass of snails contained within each respiration chamber was recorded.

The oxygen concentration was plotted as a function of time and represented by a linear regression line. The oxygen consumption rate (mg O_2_/g/h) was calculated using the formula: *R* = *b* × *V/W*, where *b* is the slope of the regression line, *V* is the water volume in the respiration chamber (L), and *W* is the wet mass of snails in respiration chamber (g). The ammonia excretion rate (μg/g/h) was determined using the equation: *E* = (*N*_t_ − *N*_0_) × *V* ÷ (*m* × *t*), where *N*t and *N*_0_ are the NH^4+^-N concentrations in the experimental chamber and blank control chamber, respectively, at the end of the experiment (μg/L), *V* is the water volume in respiration chamber (L), *W* is the wet mass of snails in respiration chamber (g), *t* is the experiment duration (h).

### 2.4. Assay of Enzymatic Activities

The tissue samples extracted from juvenile snails in each group were homogenized in an ice bath using 0.01 M phosphate-buffered saline (PBS, pH 7.4). The homogenates were then centrifuged at 12,000 r/min for 10 min at 4 °C to obtain the supernatants, which were stored in the same buffer at 4 °C for subsequent enzymatic activity analyses. The activities of pyruvate kinase (PK), lactate dehydrogenase (LDH), acid phosphatase (ACP), alkaline phosphatase (AKP), total superoxide dismutase (SOD) activity, and total protein (TP) content were measured using a microplate reader (SPARK 10 M, Grodig, Austria) with assay kits provided by Suzhou Grace Biotechnology Co., Ltd., Suzhou, China, following the manufacturer’s instructions.

Pyruvate kinase catalyzes the production of pyruvate from Phosphoenolpyruvate (PEP) in the presence of ADP, which is converted to lactate by LDH and to NADH by NADH. The enzyme activity was calculated by measuring the absorbance at 520 nm. Lactate dehydrogenase catalyzes the formation of pyruvate from lactic acid, which reacts with 2,4-dinitrophenylhydrazine to form dinitrophenylhydrazone pyruvate, which is brownish-red in alkaline solution. The enzyme activity was calculated by measuring the absorbance at 440 nm. Acid phosphatase and alkaline phosphatase activities were calculated according to the principle that phenol interacts with 4-aminoantipyrine in alkaline solution to form red quinone derivatives by oxidation with potassium ferricyanide, and the absorbance of quinone derivatives was measured at 520 nm. To measure the SOD activity, 30 μL of supernatant were mixed with 1.3 mL of reaction solution containing 0.75 mM NBT and 20 mM riboflavin. The mixture was incubated at 37 °C for 40 min and measured for the absorbance at 550 nm. One unit of SOD was defined as the activity amount that resulted in 50% inhibition of the production of anion free radicals in 1 mL of reaction solution. All results obtained were standardized based on TP levels.

### 2.5. Assay of Gene Expression of Enzymes

The total RNA from juvenile snail tissue samples in each group was extracted using kits (Biospin, Hangzhou, China) following the manufacturer’s protocol. RNA degradation and contamination were assessed by 1% agarose gel electrophoresis. The concentration and integrity of the RNA were measured using a NanoDrop ND-2000C spectrophotometer (Thermo, Waltham, MA, USA). Total cDNA synthesis was performed with Prime Script™ RT Master Mix kits (TaKaRa, Dalian, China) in a final volume of 20 μL according to the manufacturer’s instructions. Quantitative PCR (qPCR) was conducted on a QuantStudio Real-Time PCR system with a total reaction volume of 20 μL utilizing an Easy™ Mix-SYBR Green I kit (Foregene, Chengdu, China), adhering to the following thermal cycling program: initial denaturation at 95 °C for 5 min; followed by 35 cycles of amplification consisting of denaturation at 95 °C for 30 s, annealing at 55 °C for 30 s, and extension at 72 °C for one minute. In order to obtain the partial sequences of *PK*, *LDH*, *ACP*, *AKP*, and *SOD* genes, the tissue sample of a snail was taken and a De novo assembly transcriptome analysis was conducted. The partial sequences of the target genes are provided in the Appendix A. The primer sequences were designed using Primer Premier 5 software and verified by PCR amplification and gel electrophoresis (Table 1). *β-actin* was used as a reference gene [27]. Gene expression levels were analyzed employing the 2^−ΔΔCT^ method with β-actin serving as an internal control for qPCR analysis.

### 2.6. Statistical Analysis

Except for the data on mortality, the data on metabolic rates, enzyme activities, and gene expression are presented as medians and scatter plots. Kaplan–Meier survival curve analysis and non-parametric two-way analysis were performed for the results using the aligned-rank transform (ART) ANOVA with R software (v4.3.0, R Development Core Team 2019). Post hoc comparisons for interaction decomposition were performed using the Holm method through a simple effect test. A *p*-value less than 0.05 was considered to be statistically significant.

## 3. Results

### 3.1. Mortality and Metabolic Rates of B. areolata During Prolonged Hypoxia

Mortality in the *B. areolata* group exposed to 2.0 mg O_2_/L occurred starting at 72 h post-stress (Figure 1). With the extension of hypoxia stress, the mortality gradually increased, with a cumulative mortality rate of 5% by the end of the experiment. In contrast, for the 0.5 mg O_2_/L group, mortality was observed from 48 h onward and was significantly higher than the 2.0 mg O_2_/L group, resulting in a cumulative mortality rate of 51.67% by the end of the experiment. No deaths were recorded in the control group.

Compared to the control group, the oxygen consumption rate of *B. areolata* was significantly lower in both the 2 mg O_2_/L and 0.5 mg O_2_/L treatment groups (*p* = 0.000 and 0.000, respectively) (Figure 2). Throughout the experimental period, no significant fluctuations were observed in the oxygen consumption rate of the 2 mg O_2_/L group. In contrast, the oxygen consumption rate of the 0.5 mg O_2_/L group exhibited an initial decrease, followed by an increase, and subsequently a significant reduction (*p* = 0.000). The oxygen consumption rate was significantly lower than that of the 2 mg O_2_/L group at time points of 0 h, 48 h, 120 h, and 144 h (*p* = 0.000, 0.004, 0.015, and 0.001, respectively). Two-way analysis showed there was an interaction between the intensity and duration of hypoxic stress.

Compared to the control group, the ammonia excretion rate in the 2 mg O_2_/L group exhibited significant fluctuations and was significantly lower (*p* < 0.05) for most of the duration observed (Figure 3). The ammonia excretion rate in the 0.5 mg O_2_/L group first lower and then increased; it remained significantly lower than that of the control group (*p* < 0.05) and was also significantly lower than that of the 2 mg O_2_/L group at time points of 0 h, 24 h, and 120 h (*p* = 0.01, 0.000, and 0.000, respectively). Interaction between the intensity and duration of hypoxic stress was observed.

### 3.2. Biochemical Changes in B. areolata During Prolonged Hypoxia Stress

As shown in Figure 4, the PK activity of *B. areolata* in the control group remained stable, except for a significant decrease at 96 h (*p* = 0.023). Compared to the control group, the PK activity of *B. areolata* in the 2 mg O_2_/L treatment group gradually increased, ultimately reaching its maximum value at 120 h. The PK activity in the 0.5 mg O_2_/L group displayed a similar trend, peaking at 144 h and being significantly higher (*p* = 0.001) than that of the 2 mg/L group at 48 h; conversely, it was significantly lower (*p* < 0.05) than that of the 2 mg/L group at 120 h. In comparison with the control group, the LDH activity in the 2 mg/L treatment remained stable during initial hypoxic stress but showed a significant decrease after 48 h (*p* = 0.000) (Figure 4). In contrast, LDH activity in the 0.5 mg/L group was significantly higher (*p* < 0.05) at 0 h but subsequently demonstrated a marked decline compared to the control group. LDH activity in the 0.5 mg O_2_/L group was significantly lower than that of the 2 mg/L group at 24 h, 72 h, and 144 h (*p* = 0.01, 0.003, and 0.003, respectively), and significantly higher at 0 h and 96 h (*p* = 0.016, and 0.000, respectively).

The ACP activity of *B. areolata* in the 2 mg O_2_/L group exhibited a gradual decline, which was significantly lower than that observed in the control group after 48 h (*p* < 0.05), reaching its minimum at 144 h. The ACP activity in the 0.5 mg O_2_/L group showed a significant decline (*p* = 0.000) after 24 h compared to the control group. The ACP activity was significantly lower in the 0.5 mg O_2_/L group than that in the 2 mg O_2_/L group at 48 h and 120 h (*p* = 0.000 and 0.031, respectively). The AKP activity in the 2 mg/L group significantly fluctuated, and was higher at 24 h, 48 h, 96 h, and 120 h compared with the control group (*p* = 0.017, 0.019, 0.006, and 0.009, respectively) (Figure 4). The AKP activity in the 0.5 mg O_2_/L group was significantly higher than that of the control group at 96 h and 120 h (*p* = 0.011 and 0.003, respectively), and significantly lower than that of the 2 mg O_2_/L group at 144 h (*p* = 0.016). The SOD activity decreased significantly (*p* < 0.05) under hypoxic stress (Figure 4). With the extension of the stress, the SOD activity of the 2 mg O_2_/L group gradually increased, and that of the 0.5 mg O_2_/L group reached the lowest value at 48 h and then gradually increased. Compared with the 2 mg O_2_/L group, the SOD activity was significantly higher in the 0.5 mg O_2_/L group at 96 h (*p* = 0.000), and significantly lower at 48 h and 144 h (*p* = 0.000, and 0.004, respectively). Two-way analysis showed that for all the enzymatic activities, there was an interaction between the intensity and duration of hypoxic stress.

### 3.3. Expression of Related Enzyme Genes of B. areolata Under Hypoxia Stress

Compared to the control group, the expression of the *PK* gene in *B. areolata* was significantly downregulated in the 2 mg O_2_/L treatment group at both 0 h and 72 h (*p* = 0.001 and 0.024, respectively); however, it exhibited a significant upregulation at 144 h (Figure 5). In contrast, *PK* gene expression in the 0.5 mg O_2_/L group showed a significant increase at 144 h (*p* = 0.036). Except at 24 h and 120 h, the *PK* gene expression level in the 0.5 mg O_2_/L group was higher than that in the 2 mg O_2_/L group. Compared to the control group, *LDH* gene expression in the 2 mg O_2_/L treatment group was significantly downregulated at 24 h (*p* = 0.002); however, it exhibited a significant upregulation at both 96 h and 144 h (*p* = 0.007 and 0.013, respectively) (Figure 5). *LDH* gene expression in the 0.5 mg O_2_/L group showed a significant upregulation (*p* < 0.05) at both 0 h and 120 h but was significantly downregulated (*p* < 0.05) at other time points. The *LDH* gene expression level in the 0.5 mg O_2_/L group was significantly higher than that in the 2 mg O_2_/L group at both 0 h and 120 h (*p* = 0.024 and 0.000, respectively), while it was significantly lower at other time points (*p* < 0.05).

The expression level of the *ACP* gene in *B. areolata* at 2 mg O_2_/L was significantly lower than that in the control group after 24 h (*p* < 0.05) (Figure 5). The *ACP* gene expression level in the 0.5 mg O_2_/L group was significantly reduced compared to the control group (*p* < 0.05), and it was also significantly lower than that of the 2 mg O_2_/L group at 48 h, 72 h, and 144 h (*p* = 0.018, 0.027, and 0.048, respectively). The expression level of the *AKP* gene in the 2 mg O_2_/L group did not show a significant difference compared to the control group (Figure 5). In contrast, the *AKP* gene expression level in the 0.5 mg O_2_/L group was significantly higher than that of the control group at most time points (*p* < 0.05). Compared with the control group, *SOD* gene expression in the 2 mg O_2_/L group was significantly downregulated (*p* < 0.05), except at 48 h (Figure 5). Similarly, *SOD* gene expression in the 0.5 mg/L group exhibited significant downregulation (*p* < 0.05), except at 0 h and 24 h. No significant differences were observed between the 2 and 0.5 mg O_2_/L groups except at 48 h.

## 4. Discussion

### 4.1. Effects of Prolonged Hypoxic Stress on Mortality and Metabolism of B. areolata

In the present study, we elucidated the significant effects of prolonged hypoxic stress on the survival and physiological responses of juvenile *B. areolata*. The findings indicated that at a lower dissolved oxygen level (0.5 mg/L), the juvenile *B. areolata* presented a significantly higher mortality rate. Our previous acute study suggests that 2 mg O_2_/L may represent the threshold for *B. areolata* to sustain aerobic metabolism; furthermore, mortality appears to occur when dissolved oxygen levels fall below this threshold. It has been found in many shellfish that below a certain dissolved oxygen threshold, the mortality rose dramatically [23,28,29]. The survival rates of the scallop *Chlamys farreri* exposed to 2.0, 2.5, and 3.0 mg/L DO are close, ranging from 59% to 69%, while the survival rate declined to less than 50% after exposure to 1.5 mg O_2_/L [30]. A similar phenomenon was observed in the bivalve *Macoma balthica* as the mortality rate sharply increased when dissolved oxygen levels dropped below 2.5 mg/L [31].

In the present study, mortality was observed at 48 h and 72 h in the 0.5 mg/L and 2.0 mg/L groups, respectively, indicating that juvenile *B. areolata* exhibits a certain degree of tolerance to hypoxic stress for a limited duration. However, compared with the burrowing and attaching shellfish, such as clams and oysters, the hypoxia tolerance of juvenile *B. areolata* is weak. *Ruditapes philippinarum* has been shown to exhibit a high tolerance to hypoxia, with a 20-day LC50 value of 0.57 mg O_2_/L and a 422 h LC50 at 0.5 mg O_2_/L [28]. In contrast, the oyster *C. virginica* began to show mortality by day 7 when exposed to 2 mg O_2_/L [23]. Fu et al. (2024) reported that mortality occurred in the adult *B. areolata* (shell length: 44.56–44.90 mm) at 48 h after exposure to 0 mg O_2_/L, and survival declined to less than 50% after 144 h [25].

The high mortality observed in *B. areolata* exposed to 0.5 mg O_2_/L may be due to the insufficient energy acquisition and failure to maintain energy homeostasis. Nam et al. (2020) found that shell growth and meat accumulation in *Haliotis discus hannai* were significantly affected by low DO concentrations, and believe this is due to the decrease in metabolic rates and biochemical disruptions [29]. Another reason could be the underutilization of energy resources. Most enzymes have optimum pH [32,33]. In marine molluscs, both intracellular and extracellular pH decrease during anaerobiosis [4,34]. Acidic by-products of anaerobic metabolism could shift pH beyond some threshold, which may have adverse effects on enzyme functions and intracellular energy transduction.

Metabolic strategies under hypoxic conditions are vital for keeping an organism alive [28]. Measurements of oxygen consumption and the production of metabolic waste, such as ammonia excretion, in organisms provide an indirect approach for estimating the metabolism of aquatic animals [29]. In this study, we observed that both the oxygen consumption rate and ammonia excretion rate declined with the decreasing dissolved oxygen concentration. Furthermore, a significant decrease in both parameters was observed throughout the hypoxia exposure period. These findings suggest that hypoxic stress significantly impacts energy metabolism in *B. areolata*. Similar results have been documented in other marine shellfish, including the oyster *C. gigas*, the scallop *C. farreri*, the Manila clam *R. philippinarum*, the clam *Tegillarca granosa*, and the abalone *H. discus hannai* [10,28,29,30,35]. When confronted with hypoxia, these shellfish exhibited reduced respiration and metabolic rates.

Lowering the metabolic rate is one of the most important strategies employed by shellfish to cope with hypoxic stress. By reducing metabolic rates through anaerobic metabolism, the rates of all ATP-utilizing reactions in cells are strongly suppressed to a level that matches the anoxic rate of ATP production to survive hypoxic stress [4]. According to Pörtner and Farrell [36], when organisms are exposed to extreme stress conditions, anaerobic metabolism is activated upon reaching a critical threshold of the stressors. Anaerobic respiration produces only 2 ATP molecules per molecule of glucose degraded, which is significantly lower than the 32 ATP molecules generated through the complete oxidation of glucose [37]. Liu et al. [38] observed that following exposure to hypoxia (1.5 mg O_2_/L), the concentrations of anaerobic metabolites, such as succinate, lactate, acetate, fumarate, and propionate, in the subtidal gastropod *Nassarius conoidalis* increased markedly. The significant decline in oxygen consumption rate indicates that *B. areolata* may shift, at least partially, towards an anaerobic pathway under conditions of hypoxic stress. This can be seen from the significant increase in LDH activity.

### 4.2. Effects of Prolonged Hypoxic Stress on Biochemical Profiles and Gene Expression of B. areolata

Energy metabolism is accomplished through a series of enzymatic reactions that are influenced by the activity and expression levels of the involved enzymes. PK serves as a key regulatory enzyme in glycolysis. In juvenile *B. areolata*, the significant increase in PK activity and gene expression levels indicates the elevated glycolytic rate during hypoxic conditions. Similarly, *R. philippinarum* exposed to 0.5 mg O_2_/L exhibited a significant increase in PK activity [28]. In the clam *T. granosa*, PK activity exhibited a significant increase when the dissolved oxygen concentration decreased to 0.5 mg/L [10]. Conversely, Moullac et al. [39] reported a notable reduction in both PK activity and gene expression in the oyster *C. virginica* when oxygen levels were lowered to 1.96 mg/L for a duration of 20 days. This discrepancy may be attributed to species-specific differences or could stem from variations in hypoxic stress treatments or responsive mechanisms; indeed, in this study, both PK enzyme activity and *PK* gene expression levels demonstrated considerable fluctuations during periods of hypoxic stress.

LDH is also a key enzyme in the anaerobic metabolic pathway, catalyzing the conversion of pyruvate to lactate for ATP production. In the present study, the significant increase in LDH activity observed in *B. areolata* exposed to 0.5 mg O_2_/L (0 h) indicates that hypoxic stress triggers anaerobic metabolism. Similarly, an increase in LDH activity was observed in the gills of *R. philippinarum* after exposure to 2 mg O_2_/L; however, a different trend was observed in the hepatopancreas [19]. Li et al. (2019) reported that LDH activity in the adductor muscle of *R. philippinarum* exhibited no significant change at 2 mg O_2_/L on day 10 [28]; however, it was significantly elevated compared to the control group on day 20. These findings suggest that both the sampled tissues and the timing of measurements may influence the results. It has been established that shellfish possess multiple pathways for energy metabolism, including the glucose-succinate pathway, pyruvate-lactate pathway, proline pathway, and opine pathway [4]. In the present study, *B. areolata* may utilize the pyruvate-lactate pathway as its primary respiratory metabolism pathway during the early stages of stress. To mitigate the effects of acid accumulation on cellular homeostasis in vivo during prolonged hypoxia, *B. areolata* may shift to alternative metabolic pathways that yield volatile or less acidic end products instead of lactate, thereby resulting in a decrease in LDH activity. Wang et al. [10] also observed a reduction in lactate content in the clam *T. granosa* following exposure to hypoxic stress. In the present study, the variation in LDH activities did not fully align with gene expression patterns, as the peak values were observed at different time points. This finding underscores the complexity of metabolic regulation in *B. areolata* under hypoxic stress. While there is generally a correlation between gene expression and phenotypic traits, this relationship does not reflect a direct one-to-one correspondence. Post-translational modifications of proteins, such as phosphorylation, acetylation, ubiquitination, methylation, and glycosylation, also play a significant role in determining the final phenotype [40,41].

Studies have demonstrated that hypoxia can physiologically and pathologically modulate immune activity in cells [42]. ACP serves as a marker enzyme for lysosomes and plays a crucial role in the killing and digestion of microbial pathogens during immune responses. Its activity can serve as an indicator of the immune status in molluscs [43]. In the scallop, lower ACP activity was detected in the disease or stress state than in the healthy state [44]. In the present study, the ACP activity and *ACP* gene expression in *B. areolata* significantly decreased following exposure to hypoxic stresses of 2 mg O_2_/L and 0.5 mg O_2_/L. This finding suggests that hypoxic stress inhibits the activity of hydrolytic enzymes involved in cellular immunization processes within *B. areolata*, which may be detrimental to the survival of juvenile individuals of *B. areolata*. In the scallop *C. farreri*, the ACP activity in the haemolymph significantly decreased 14 days after dissolved oxygen levels fell to 2.5 mg O_2_/L [43]. This reduction in ACP activity and *ACP* gene expression is likely a consequence of metabolic arrest induced by anaerobiosis, as well as alterations in lysosomal integrity resulting from hypoxic stress [43]. AKP is a multifunctional enzyme that plays a crucial role in phosphate metabolism and immune defense, including detoxification of lipopolysaccharides, and it functions as a transphosphorylase at alkaline pH [15,43]. In this study, the AKP activity and gene expression were found to be elevated after hypoxic stress, particularly in the group exposed to 0.5 mg O_2_/L. Similarly, AKP activity in the hepatopancreas of *R. philippinarum* significantly increased after five days of hypoxic stress at a concentration of 2.0 mg O_2_/L [19]. In this study, the ACP activity exhibited an opposite trend compared to that of AKP. Comparable findings have been reported in the scallop *C. farreri* [43]. The optimal pH for the functioning of these two enzymes differs; therefore, further investigations are warranted to ascertain whether the pH environment within *B. areolata* cells is altered as a result of anaerobic metabolism.

It has been demonstrated that reactive oxygen species (ROS), including superoxide, hydrogen peroxide, and hydroxyl radicals, are generated in cells under hypoxic conditions [16,45]. SOD plays a crucial role in protecting cells from ROS-induced damage and serves as an important biomarker for assessing immune responses in shellfish. In the present study, we observed a significant reduction in both SOD activity and gene expression under hypoxia. These findings indicate a possible inhibition of antioxidative responses in the snails during exposure to low levels of DO, which has been observed in other aquatic animals [30,46]. The decrease in metabolic rates might also lead to the reduced SOD activity through suppressing cell respiration and superoxide ion production. Reductions in SOD activities have also been observed in the marine gastropod *Littorina littorea* after 6 days of exposure to anoxic conditions [47]. The triangle sail mussel, *Hyriopsis cumingii*, exhibited a significant reduction in SOD activity under hypoxic conditions of 1 mg O_2_/L and 3 mg O_2_/L [45]. Similarly, a notable decrease in SOD activity was observed in the abalone *H. discus hannai* after two months of exposure to hypoxia at 2.5 mg O_2_/L [19]. In contrast, no significant changes were detected in SOD activity or *SOD* gene expression in the mussel *M. galloprovincialis* following 24 h of stress at 2 mg O_2_/L [48]. Conversely, the clam *T. granosa* demonstrated a significantly elevated SOD activity when subjected to hypoxic conditions of 0.5 mg O_2_/L [10]. These findings indicate that antioxidant defense to hypoxia varies considerably among different species.

## 5. Conclusions

The present study demonstrated that exposure to 2 mg O_2_/L and 0.5 mg O_2_/L significantly suppressed the metabolic activities and biochemical functions of *B. areolata*. Under long-term exposure to hypoxia, *B. areolata* failed to maintain energy homeostasis and suffered biochemical disruptions, leading to increased mortality. The present study also highlights the importance of maintaining appropriate dissolved oxygen concentrations during the culture and transportation of *B. areolate*. These findings contribute to our understanding of the adaptive mechanisms employed by marine gastropods for hypoxic tolerance and provide valuable insights for minimizing hypoxic damage to *B. areolata*.

## Figures and Tables

**Figure 1 biology-14-00727-f001:**
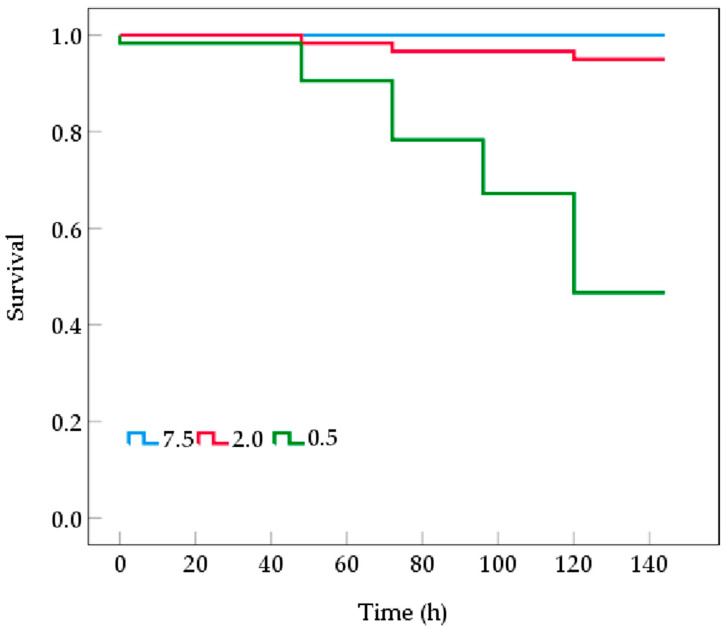
Kaplan–Meier survival curves of *Babylonia areolata* exposed to different concentrations of dissolved oxygen (0.5, 2.0, and 7.5 mg O_2_/L) (*n* = 180).

**Figure 2 biology-14-00727-f002:**
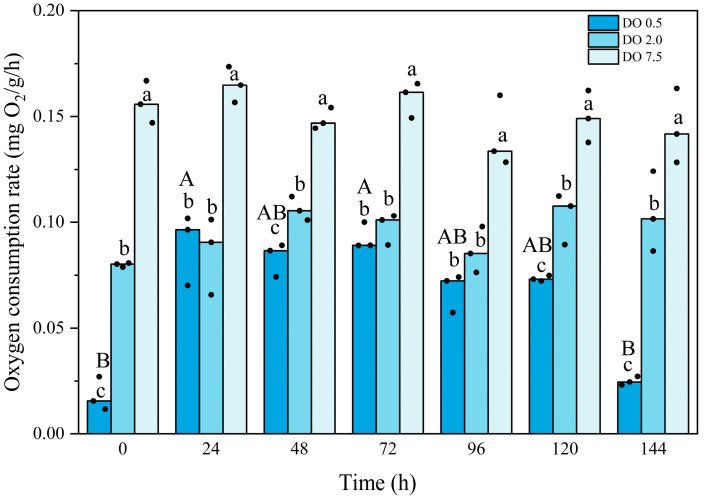
Oxygen consumption rate of *Babylonia areolata* exposed to different concentrations of dissolved oxygen (0.5, 2.0, and 7.5 mg O_2_/L) (*n* = 3). The figure is presented in medians and raw data points. Non-parametric two-way analysis and post hoc comparisons were applied. Values with different lowercase letters are significantly different between 0.5, 2.0, and 7.5 mg O_2_/L at the same time point. Different uppercase letters indicate that the values of each treatment group are significantly different at different time points.

**Figure 3 biology-14-00727-f003:**
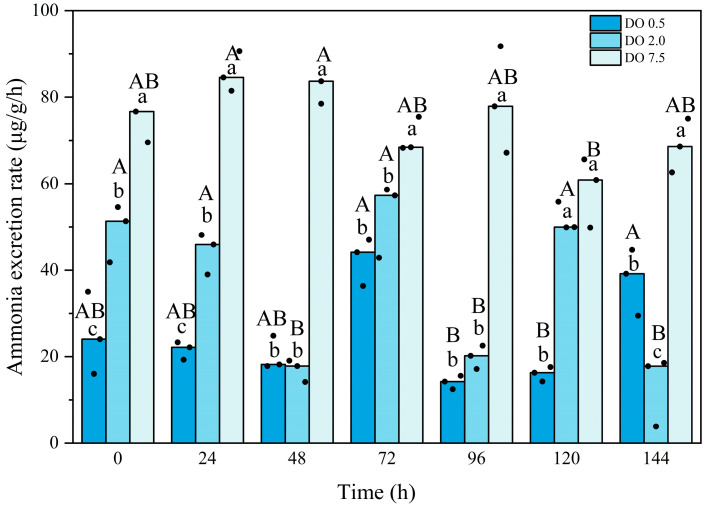
Ammonia excretion rate of *Babylonia areolata* exposed to different concentrations of dissolved oxygen (0.5, 2.0, and 7.5 mg O_2_/L) (*n* = 3). The figure is presented in medians and raw data points. Non-parametric two-way analysis and post hoc comparisons were applied. Values with different lowercase letters are significantly different between 0.5, 2.0, and 7.5 mg O_2_/L at the same time point. Different uppercase letters indicate that the values of each treatment group are significantly different at different time points.

**Figure 4 biology-14-00727-f004:**
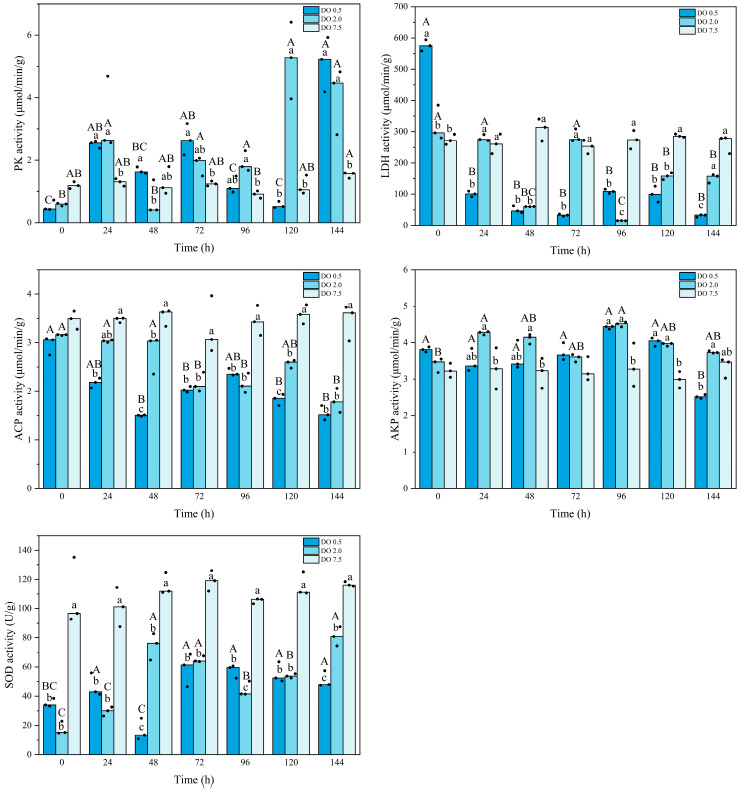
Metabolic and immune enzyme activities of *Babylonia areolata* exposed to different concentrations of dissolved oxygen (0.5, 2.0, and 7.5 mg O_2_/L) (*n* = 3). The figure is presented in medians and raw data points. Non-parametric two-way analysis and post hoc comparisons were applied. Values with different lowercase letters are significantly different between 0.5, 2.0, and 7.5 mg O_2_/L at the same time point. Different uppercase letters indicate that the values of each treatment group are significantly different at different time points.

**Figure 5 biology-14-00727-f005:**
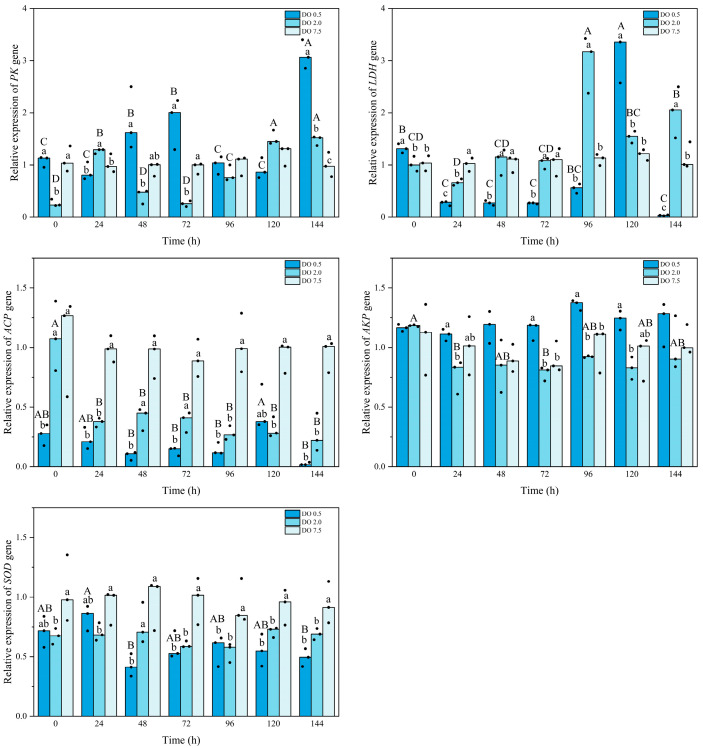
Relative gene expression of metabolic and immune enzymes of *Babylonia areolata* exposed to different concentrations of dissolved oxygen (0.5, 2.0, and 7.5 mg O_2_/L) (*n* = 3). The figure is presented in medians and raw data points. Non-parametric two-way analysis and post hoc comparisons were applied. Values with different lowercase letters are significantly different between 0.5, 2.0, and 7.5 mg O_2_/L at the same time point. Different uppercase letters indicate that the values of each treatment group are significantly different at different time points.

**Table 1 biology-14-00727-t001:** Primer sequences and internal reference used for qPCR.

Gene Names	Forward/Reverse Sequence (5′ to 3′)
*LDH*-F	GAGGTCGAGTCTTGGTCGTT
*LDH*-R	ACCGCTCTGCCAGTCTTCA
*PK*-F	GCATTTGTGCCATCTTGTA
*PK*-R	GCCATACCGTGTCCTCTAC
*SOD*-F	TGCCAAGGTCACATCAATC
*SOD*-R	ATGCCTACCGCACTCGTTT
*ACP*-F	AGCGTAGACACTGCTCGTA
*ACP*-R	GATGCTGGGAAACTGGGAC
*AKP*-F	GTTGTTGCTGGTAAAGATGA
*AKP*-R	CAAGTTTGGGCTGATGAAG
*β-actin*-F	GGTTCACCATCCCTCAAGTACCC
*β-actin*-R	GGGTCATCTTTTCACGGTTGG

## Data Availability

Data is contained within the article or Appendix A.

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
