# Peer review of "Metabolic and Biochemical Responses of Juvenile *Babylonia areolata* to Hypoxia Stress"

_biology, 2025, doi:10.3390/biology14060727_

Round 1

Reviewer 1 Report

Comments and Suggestions for Authors

I'm not a specialist in gastropods, so I could fully evaluate only the methodological part of the study. Overall, I found the manuscript well written, logically organized and, as far as I could assess, valuable to the study area. I also appreciated good description of the experimental design and easily readable language. However, I have significant concerns for the current version of the manuscript, mostly with data presentation and analysis:

1) Poor presentation of the mortality data. Animal survival is the cornerstone information for this type of studies and mentioned several times in different parts of the manuscript. Yet, it is presented only in the text without much details. To my opinion, the mortality data must be presented in dynamics on a dedicated figure and described in a separate section of Results (those data are not even mentioned in the section title now).
My impression from the the experimental design is that authors didn't install separate drums to monitor mortality over time in parallel to the laboratory measurements. So, the total number of animals decreased each day, making it more difficult to track the overall survival. If that's so, the percents currently presented at the beginning of the Results are difficult to interpret without further clarification how they were calculated, and it makes better presentation of the data even more important.
In this case, the convenient way to draw the mortality data will be using the Kaplan–Meier estimator and indicating so-called right censoring for the specific individuals extracted from the experiment for the laboratory measurements while they were still alive. I would suggest the R packages "survival" and "survminer" for this purpose (examples can be found in the section Vignettes here: https://cran.r-project.org/web/packages/survival/index.html).

2) Non-transparent raw data. First of all, it is not clear how many measurements were made per each sample group both from the figures and the Methods section. There is, of course, information on the amount of taken individuals per each analysis in the experimental design, but some measurements could fail. Furthermore, raw measurement data are also absent. I suggest:
- Clearly stating "n =" in each figure caption or indicating the sample size in other ways (mandatory).
- Re-drawing all the figures in a way more friendly to presenting data distributions. Boxplots are probably not the best alternative since (as far as I got from the experimental design) the group size for "enzyme activity and gene expression analyses" could be no more than six, while boxplots and analogs are better with n > 10. Probably, medians (could be presented as bars similarly to means) and raw data points are the best option here. Some details can be found in dedicated guidelines such as this one: https://journals.plos.org/plosbiology/article?id=10.1371/journal.pbio.1002128. Otherwise, the authors must give a clear statistical explanation why data with such a small group size can still be presented as means ± sd.
- Depositing the raw data as the supplementary material or in some repository (GitHub, Zotero, etc.), for example, as simple Excel spreadsheet.

3) Current description of statistical analysis is unacceptably scarce and makes me question the presented results. I can conclude from the experimental design that groups included at most 12 replica (or 4 individuals were used as one replica for metabolic rate and ammonia?) for some parameters, while the majority of parameters included only six individuals. This is far from enough for the tests checking the data normality and homoscedasticity (at the very least 15-20 according to most guidelines). I.e., the data could indeed pass the Levene and Shapiro-Wilk tests, but those could not be applied at the first place. Thus, the authors must either explain why they used parametric statistics instead of non-parametric tests (maybe there is some a priori information about normality of the measured parameters obtained for a large group of the studied object?) or switch to non-parametric tests. There are also some related issues:
- Even with parametric tests there should be more details about the applied post-hoc test, the type of correction for multiple comparisons and also the explanation how the corrections were applied (there were probably only comparisons within each timepoint, but how many comparisons were processed within each correction, 2 or 14?).
- Information like "(p < 0.05)" along the text of Results is meaningless. I appreciate highlighting the specific comparisons with statistically significant differences, but specific p-values should be indicated if the authors decided to do so.
- L186, what was the purpose of ANOVA? To compare groups at each time point or dynamics within each group?

4) I'm highly concerned with the term "immune response" in title and later along the text. Usually the term is related to specific markers like levels of pattern recognition proteins, hemocyte parameters, etc., not activities of pretty general enzymes measured here from the whole body (i.e., not tissues related to immune system). The link [31] on L379 for ACP or stating that "AKP is a multifunctional enzyme" on L388 don't help much in justifying their status as immune enzymes. The authors must clearly explain their use of terms in Introduction or reconsider the status of ACP and AKP within the study.

5) The manuscript suffers from mostly just presenting the data and mentioning some literature facts rather than deeply analyzing them in combination. In particular:
- Abstract is overwhelmed with facts, while the conclusion "results indicated that B. areolate may respond to hypoxic stress by reducing its metabolic rate and modulating the expression of metabolism related enzyme" is obvious and the conclusion "hypoxic stress was found to significantly impact the antioxidant capacity and immune function" is just not correct (besides immune response, antioxidant capacity was not measured here; SOD is just one of many markers).
- the Introduction lacks clear testable hypotheses
- Results could benefit from focusing on dynamics of changes, rather than simply listing significant differences (although I acknowledge the right of authors to describe results in the way convenient for them)
- Most importantly, Discussion finally places the data into some context, but doesn't produce many novel conclusions about biology of the species (my personal complaint here would also be the current discussion of the data parameter-by-parameter rather than converging the results from different parameters)
- L417-419 there are no data in the manuscript supporting this conclusion
- L420-422 and this conclusion is obvious from basic physiology and biochemistry.

I suggest building Discussion+Conclusions (and probably the end of Introduction) from actual mortality data and their comparison to mortality of other species. The authors already state this nice proposition that "This difference in tolerance may be attributed to the relatively weak mobility of oysters and clams, which have evolved mechanisms to withstand low oxygen conditions; whereas B. areolate, being more mobile, can migrate towards more suitable habitats". This is of course to be expected from the species ecology, but, as far as I know, the actual degree of variability in sensitivity to hypoxia between different species depending on their ecological niche is yet to be determined for ecophysiology of different taxonomical groups. Thus, I would suggest focusing around the search for the biochemical changes that explain the interspecific differences in tolerance to not just hypoxia generally, but specific low levels of oxygen along the whole Discussion.

More specific issues and questions:
- how Babylonia areolate is related to Babylonia areolata?
- L79 mentioning PK and LDH is good, but probably other markers also deserve some introduction here.
- L83, please mention the author and year of the species description as requested in the International Code of Zoological Nomenclature.
- L93, I don't see citation to a recent study on Babylonia areolata in hypoxic conditions along the whole manuscript: https://www.sciencedirect.com/science/article/pii/S2352513424002199. Perhaps, this is a good place to mention this and potentially other relevant studies?
- L104 what about feeding during the experiment?
- L96 why juveniles were chosen? Are they transported alive most often?
- L120 at this point the 2 mg/ml group was already at this conditions for an hour, is that right? Or the starting time points were different in each group?
- L121 what was the motivation for the experiment duration? Probably known time usually used for transportation?
- L122, "performed every 72 hours", so just once during the main experiment?
- L127 were the individuals used for respiration measurements later returned to the same aquaria?
- L128 was each snail used for both enzyme activity and gene expression analyses? I could not find it later in Methods
- L135 so, for each measurement the chamber contained 4 snails from the same drum? Did the authors use one measurement setup and had to process all replicas sequentially? If so, how could the starting point 0 h be obtained with 1-h-long measurements?
- L164 I could not find the protocols on the web-pages related to Suzhou Grace Biotechnology Co. Authors might consider briefly mentioning here at least the biochemical principles behind each protocol.
- L180 what was the source of sequences for designing the primers? I.e. how reliable are they?
- L202 "a significant reduction (p < 0.05)" I see two-fold decrease, yes, but technically the dynamics within each group was not considered during the statistical analysis, was it? So, the 120 and 144 h groups at 0.5 mg/ml were not tested, am I right? Considering small sample sizes, the difference could be not significant.
- L294 "Our previous acute experiment" could you please provide the reference here or, even better, in the Introduction?
- L298 "These findings suggest that 2 mg/L may represent the threshold" it is not at all clear if those are referenced information or some kind of mix between a previous study and the current one
- L315 "generally negatively correlated", the term "correlated" usually requires a correlation coefficient; while earlier in the text it was obvious what result would give such a coefficient being calculated (so I didn't object), here the word "generally" is unacceptable. Please provide the specific coefficients in the Results or remove the term "correlation" here.
- L333 "means to conserve energy" or inability to use aerobic oxidation?
- L335 maybe your data on LDH should be mentioned here?
- L342 "despite overall metabolic depression" I don't see a contradiction here; what other energy sources could be used? "overall metabolic depression" in L342 mostly represents the aerobic part.
- L369-371 this conclusion is drawn purely from one timepoint for LDH expression, namely 120 h. Could it be a measurement error? Again, what was the sample size here?
- L382-383 "inhibits the activity of hydrolytic enzymes involved in cellular immunization processes" can the enzyme be involved into hydrolysis of own proteins, not only foreign ones?
- L399 this is the first line where ROS are mentioned and the role of SOD is explained. Again, probably should be done in Introduction.

Author Response

Thank you very much for taking the time to review this manuscript and giving insightful comments. Please find the detailed responses below and the corresponding revisions in track changes in the re-submitted files.

  1. Responses for general evaluation

Comments 1: Poor presentation of the mortality data. Animal survival is the cornerstone information for this type of studies and mentioned several times in different parts of the manuscript. Yet, it is presented only in the text without much details. To my opinion, the mortality data must be presented in dynamics on a dedicated figure and described in a separate section of Results (those data are not even mentioned in the section title now).
  My impression from the experimental design is that authors didn't install separate drums to monitor mortality over time in parallel to the laboratory measurements. So, the total number of animals decreased each day, making it more difficult to track the overall survival. If that's so, the percents currently presented at the beginning of the Results are difficult to interpret without further clarification how they were calculated, and it makes better presentation of the data even more important.
  In this case, the convenient way to draw the mortality data will be using the Kaplan–Meier estimator and indicating so-called right censoring for the specific individuals extracted from the experiment for the laboratory measurements while they were still alive. I would suggest the R packages "survival" and "survminer" for this purpose (examples can be found in the section Vignettes here: https://cran.r-project.org/web/packages/survival/index.html).

Response 1: Thank you for the helpful suggestion. We agree with the reviewer that mortality is indeed important data. Our original purpose was to measure physiological and biochemical indicators of the snails under hypoxic stress, so we did not set up a separate drum to monitor mortality. Because the juvenile snails are relatively small, it is difficult to determine whether they are dead by visual observation. Usually we pull their siphons with tweezers to observe their response, which may cause operational duress to some extent. Therefore, we counted the mortality rate at each time point of metabolic rate measurement and tissue sampling. Since the specific time of death of each individual was not recorded, it was not possible to draw survival curves with the Kaplan-Meier method. A line plot was provided (Figure 1). As stated by the reviewer, because samples were taken every day, the total number of snails decreased each day, and the cumulative mortality rate was calculated assuming that the sampled individuals were still alive. In our opinion, the effect of hypoxic stress on the survival of juvenile snails can also be estimated by the mortality data at fixed time points, and will not have a major impact on the results of the manuscript.

Comments 2: Non-transparent raw data. First of all, it is not clear how many measurements were made per each sample group both from the figures and the Methods section. There is, of course, information on the amount of taken individuals per each analysis in the experimental design, but some measurements could fail. Furthermore, raw measurement data are also absent. I suggest:
    - Clearly stating "n =" in each figure caption or indicating the sample size in other ways (mandatory).
    - Re-drawing all the figures in a way more friendly to presenting data distributions. Boxplots are probably not the best alternative since (as far as I got from the experimental design) the group size for "enzyme activity and gene expression analyses" could be no more than six, while boxplots and analogs are better with n > 10. Probably, medians (could be presented as bars similarly to means) and raw data points are the best option here. Some details can be found in dedicated guidelines such as this one: https://journals.plos.org/plosbiology/article?id=10.1371/journal.pbio.1002128.

    Otherwise, the authors must give a clear statistical explanation why data with such a small group size can still be presented as means ± sd.
    - Depositing the raw data as the supplementary material or in some repository (GitHub, Zotero, etc.), for example, as simple Excel spreadsheet.

Response 2: (1) All figures have been redrawn as suggested by the reviewers. All figures except Figure 1 were changed to median (bar plot) + raw data points (scatter plot). “n=3” has been labeled in all figure captions; (2) raw data have been submitted as appendixes.

Comments 3: Current description of statistical analysis is unacceptably scarce and makes me question the presented results. I can conclude from the experimental design that groups included at most 12 replica (or 4 individuals were used as one replica for metabolic rate and ammonia?) for some parameters, while the majority of parameters included only six individuals. This is far from enough for the tests checking the data normality and homoscedasticity (at the very least 15-20 according to most guidelines). I.e., the data could indeed pass the Levene and Shapiro-Wilk tests, but those could not be applied at the first place. Thus, the authors must either explain why they used parametric statistics instead of non-parametric tests (maybe there is some a priori information about normality of the measured parameters obtained for a large group of the studied object?) or switch to non-parametric tests. There are also some related issues:
    - Even with parametric tests there should be more details about the applied post-hoc test, the type of correction for multiple comparisons and also the explanation how the corrections were applied (there were probably only comparisons within each timepoint, but how many comparisons were processed within each correction, 2 or 14?).
    - Information like "(p < 0.05)" along the text of Results is meaningless. I appreciate highlighting the specific comparisons with statistically significant differences, but specific p-values should be indicated if the authors decided to do so.
    - L186, what was the purpose of ANOVA? To compare groups at each time point or dynamics within each group?

Response 3: Thank you for the helpful suggestion. By seeking professional help, we redid our data analysis and statistics. All statistics have been changed to nonparametric two-way analysis using the aligned-rank transform (ART) method. The results showed interactions between intensity and duration of hypoxic stress for all indicators. Post-hoc comparisons for interaction decomposition were done using the Holm method. The p values have been added in the Results. Differences between treatments at each time point and between time points for each treatment group have also been labeled in the figures.

Comments 4: I'm highly concerned with the term "immune response" in title and later along the text. Usually the term is related to specific markers like levels of pattern recognition proteins, hemocyte parameters, etc., not activities of pretty general enzymes measured here from the whole body (i.e., not tissues related to immune system). The link [31] on L379 for ACP or stating that "AKP is a multifunctional enzyme" on L388 don't help much in justifying their status as immune enzymes. The authors must clearly explain their use of terms in Introduction or reconsider the status of ACP and AKP within the study.

Response 4: We agree with the reviewer that acid phosphatase and alkaline phosphatase are multifunctional enzymes that play important roles in the response of marine organisms to hypoxic stress, in addition to being associated with immune function in shellfish. The level of ACP and AKP activities is closely related to the intensity of stress to which marine organisms are subjected. In this study, we took whole body tissue because of the difficulty in drawing serum and distinguishing different tissues because of the small size of the juvenile snails. We have changed the title of the manuscript and explained the reasons for choosing these two enzymes in the Introduction.

Comments 5: The manuscript suffers from mostly just presenting the data and mentioning some literature facts rather than deeply analyzing them in combination. In particular:
    - Abstract is overwhelmed with facts, while the conclusion "results indicated that B. areolate may respond to hypoxic stress by reducing its metabolic rate and modulating the expression of metabolism related enzyme" is obvious and the conclusion "hypoxic stress was found to significantly impact the antioxidant capacity and immune function" is just not correct (besides immune response, antioxidant capacity was not measured here; SOD is just one of many markers).
    - the Introduction lacks clear testable hypotheses
    - Results could benefit from focusing on dynamics of changes, rather than simply listing significant differences (although I acknowledge the right of authors to describe results in the way convenient for them)
    - Most importantly, Discussion finally places the data into some context, but doesn't produce many novel conclusions about biology of the species (my personal complaint here would also be the current discussion of the data parameter-by-parameter rather than converging the results from different parameters)
    - L417-419 there are no data in the manuscript supporting this conclusion
    - L420-422 and this conclusion is obvious from basic physiology and biochemistry.

I suggest building Discussion+Conclusions (and probably the end of Introduction) from actual mortality data and their comparison to mortality of other species. The authors already state this nice proposition that "This difference in tolerance may be attributed to the relatively weak mobility of oysters and clams, which have evolved mechanisms to withstand low oxygen conditions; whereas B. areolate, being more mobile, can migrate towards more suitable habitats". This is of course to be expected from the species ecology, but, as far as I know, the actual degree of variability in sensitivity to hypoxia between different species depending on their ecological niche is yet to be determined for ecophysiology of different taxonomical groups. Thus, I would suggest focusing around the search for the biochemical changes that explain the interspecific differences in tolerance to not just hypoxia generally, but specific low levels of oxygen along the whole Discussion.

Response 5: Thanks for the constructive comments. We have reorganized the manuscript.

(1) We have rewritten the Abstract and revised the conclusions.

(2) We have revised the Introduction and add content on hypothese.

(3) The Results has been revises. We have supplemented the mortality fiture and corrected the errors in it (The cumulative mortality in the Babylonia areolata group exposed to 2.0 mg O2/L should be 5% by the end of the experiment. The previous result was the mean deviation of the mortality data).

(4) According to the reviewer’s comments, we rewrote the Discussion and Conclusion. In addition to the discussion of specific parameters, an attempt was made to discuss the adaptive processes and mechanisms of Babylonia areolata to hypoxic stress from a holistic and systemic perspective.

  1. Responses for specific comments

Comment 1: how Babylonia areolate is related to Babylonia areolata?

Response 1: We are sorry for the spelling mistake. The Latin name of the snail has been revised to Babylonia areolata.
Comment 2: L79 mentioning PK and LDH is good, but probably other markers also deserve some introduction here.

Response 2: Thanks for the comment. Information on SOD, ACP, and AKP has been added。
Comment 3: L83, please mention the author and year of the species description as requested in the International Code of Zoological Nomenclature.

Response 3: the author and year of the species description has been added.
Comment 4: L93, I don't see citation to a recent study on Babylonia areolata in hypoxic conditions along the whole manuscript: https://www.sciencedirect.com/science/article/pii/S2352513424002199. Perhaps, this is a good place to mention this and potentially other relevant studies?

Response 4: Thanks for the suggestion. We have cited the literature in our manuscript.
Comment 5: L104 what about feeding during the experiment?

Response 5: During the acclimation, the juvenile Babylonia areolata were fed fresh mackerel. In order to prevent deterioration of water quality and to minimize disturbance to the juveniles, no diet was fed during the experiment.
Comment 6:
L96 why juveniles were chosen? Are they transported alive most often?

Response 6: because these juvenile snails were acclimated and cultured in the laboratory for 30 days, during which time the juveniles have been growing and varied individually. In order to minimize the effect of individual size on the results, snails with similar sizes were selected for the experiment.
Comment 7: L120 at this point the 2 mg/ml group was already at this conditions for an hour, is that right? Or the starting time points were different in each group?

Response 7: In the present study, the DO level was adjusted to 2.0 mg/L within half an hour and subsequently to 0.5 mg/L within one hour. The time at which the DO level reached the predetermined value was recorded as 0 hours. Therefore, in the present study, the starting time points were different for DO7.5, DO2.0 and DO1.5 treatments. The interval between different groups is about 1.5 hours. Another reason is that we did not have enough oxygen electrodes to simultaneously measure nine replicates.
Comment 8: L121 what was the motivation for the experiment duration? Probably known time usually used for transportation?

Response 8: Usually the transport period of the juvenile Babylonia areolata does not exceed 3 days, but the time of hypoxia suffered during culture may be longer, so it was set to 144 hours. In the study of Fu et al. (2004), the experimental duration was also 144 hours.
Comment 9: L122, "performed every 72 hours", so just once during the main experiment?

Response 9: Because the maintenance of different dissolved oxygen is a little troublesome, we had to frequently check and adjust the volume of nitrogen and air aerated. In order to ensure the stable DO level, the water was changed only once during the experiment.
Comment 10: L127 were the individuals used for respiration measurements later returned to the same aquaria?

Response 10: The individuals used for respiration measurements were returned to the same aquaria. Actually, the snails were placed in a mesh bag, so the same set of snails was measured each time.

Comment 11: L128 was each snail used for both enzyme activity and gene expression analyses? I could not find it later in Methods

Response 11: Every 24 hours, two snails were taken from each replicate for enzyme activity and gene expression analyses, respectively.
Comment 12: L135 so, for each measurement the chamber contained 4 snails from the same drum? Did the authors use one measurement setup and had to process all replicas sequentially? If so, how could the starting point 0 h be obtained with 1-h-long measurements?

Response 12: For each measurement, the chamber contained 4 snails from the same drum. We have three setups for respiration measurement.

Comment 13: L164 I could not find the protocols on the web-pages related to Suzhou Grace Biotechnology Co. Authors might consider briefly mentioning here at least the biochemical principles behind each protocol.

Response 13: The website of the company is https://www.geruisi-bio.com/. Brief measuring methods for enzymatic activities have been added to the manuscript.

Comment 14: L180 what was the source of sequences for designing the primers? I.e. how reliable are they?

Response 14: In order to obtain the sequence information of the related genes, we performed a transcriptome analysis. After designing and synthesizing the primers, we then verified the reliability of the primers by PCR and electrophoresis.
Comment 15: L202 "a significant reduction (p < 0.05)" I see two-fold decrease, yes, but technically the dynamics within each group was not considered during the statistical analysis, was it? So, the 120 and 144 h groups at 0.5 mg/ml were not tested, am I right? Considering small sample sizes, the difference could be not significant.

Response 15: Thanks for the suggestion. All statistics have been changed to nonparametric two-way analysis. Differences between treatments at each time point and between time points for each treatment group have also been labeled in the figures.
Comment 16: L294 "Our previous acute experiment" could you please provide the reference here or, even better, in the Introduction?

Response 16: This part has been transferred to the Introduction.
Comment 17: L298 "These findings suggest that 2 mg/L may represent the threshold" it is not at all clear if those are referenced information or some kind of mix between a previous study and the current one

Response 17: This part has been revised.
Comment 18: L315 "generally negatively correlated", the term "correlated" usually requires a correlation coefficient; while earlier in the text it was obvious what result would give such a coefficient being calculated (so I didn't object), here the word "generally" is unacceptable. Please provide the specific coefficients in the Results or remove the term "correlation" here.

Response 18: We agree with the reviewer. In the present study, both the oxygen consumption rate and ammonia excretion rate declined with the decreasing dissolved oxygen concentration. The sentence has been revised to avoid ambiguity.
Comment 19: L333 "means to conserve energy" or inability to use aerobic oxidation?

Response 19: We agree with the reviewer. The sentence has been revised as “This finding suggests an enhancement in anaerobic metabolism due to inability to use aerobic oxidation”.
Comment 20: L335 maybe your data on LDH should be mentioned here?

Response 20: Thanks for the suggestion. The results of LDH have been cited here.
Comment 21: L342 "despite overall metabolic depression" I don't see a contradiction here; what other energy sources could be used? "overall metabolic depression" in L342 mostly represents the aerobic part.

Response 21: We agree with the reviewer. The sentence has been revised to avoid ambiguity.
Comment 22: L369-371 this conclusion is drawn purely from one time point for LDH expression, namely 120 h. Could it be a measurement error? Again, what was the sample size here?

Response 22: In the present study, the sample size for both enzyme activity and gene expression assays was three. Inconsistencies in enzyme activity and gene expression of LDH were observed several times at 0, 96, 120, and 144h, particularly at 96 and 120h. As suggested by the reviewer, it is also possible that this is a measurement error. However, we have found this issue in previous studies and it has also been reported in other literature. The reason and mechanism is not yet understood.

Comment 23: L382-383 "inhibits the activity of hydrolytic enzymes involved in cellular immunization processes" can the enzyme be involved into hydrolysis of own proteins, not only foreign ones?

Response 23: Thanks for the suggestion. Because the mRNA expression level of ACP also significantly decreased, we think that this might be an inhibition of ACP synthesis or activation. Indeed, in the absence of more sophisticated research methods and more data, we are not sure which factors are responsible for the ACP results.
Comment 24: L399 this is the first line where ROS are mentioned and the role of SOD is explained. Again, probably should be done in Introduction.

Response 24: Thanks for the suggestion. The information of ROS and SOD has been added in the Introduction.

Reviewer 2 Report

Comments and Suggestions for Authors

The authors assessed enzymatic and transcriptional responses of snail to hypoxia with seven days. The authors obtained expected results, namely, decrease in metabolic rate and activation of anaerobic metabolism pathways. The results are too limit!

Title must be changed. I see no immunological parameters assessed in this study. Acid phosphatase is not solely and indicator of immune function, but it plays roles in metabolism, bone health and ... also immune responses. 

L22-23: indicate the pathways.

Abstract needs a final conclusion of the observed results. Interprete them.

Introduction is Ok. But more references to previous works on the same species or related species regarding to hypoxia responses are needed.

Methods: Indicate which organ was sampled for analysis.

How did the authors measured soluble protein levels in the tissue extract?

Statistical analysis needs revision. The authors had three levels of DO and seven sampling times. Thus they must analyze the results by repeated measure two ANOVA.

What post hoc test was used for multiple comparison?

Results: Write the exact p-values in the results. Add the name of the statistical test for each figure

Discussion

This section needs extensive work. There are some mis-iterpretations. For example, the authors wrote about the lipid peroxidation, but they did not measured any indicators of lipid peroxidation (e.g. MDA and Protein carbonyl). The decrease in SOD activity is likely due to the decrease in metabolic rate, leading to less cell respiration and superoxide ion production

Author Response

Thank you very much for taking the time to review this manuscript and giving insightful comments. Please find the detailed responses below and the corresponding revisions in track changes in the re-submitted files.

Comments 1: The authors assessed enzymatic and transcriptional responses of snail to hypoxia with seven days. The authors obtained expected results, namely, decrease in metabolic rate and activation of anaerobic metabolism pathways. The results are too limit!

Response 1: Thank you for the helpful suggestion. We have performed acute stress studies previously and found that there was a significant decrease in metabolic rates, especially after the DO dropped to 2 mg O2/L. Similar results have also been reported in many shellfish. However, in our previous study, no mortality was observed. In Babylonia areolate farming or transport, the hypoxic stress usually occurs gradually and will not last for a long time. So we carried out a one-week stress experiment, and the results showed that Babylonia areolate is sensitive to hypoxic stress, with high mortality occurred.

Comments 2: Title must be changed. I see no immunological parameters assessed in this study. Acid phosphatase is not solely and indicator of immune function, but it plays roles in metabolism, bone health and ... also immune responses. 

Response 2: Thank you for the suggestion. The title has been changed to Metabolic and biochemical responses of juvenile Babylonia areolata to hypoxia stress.

Comments 3: L22-23: indicate the pathways.

Response 3: the anaerobic energy metabolic pathway has been added.

Comments 4: Abstract needs a final conclusion of the observed results. Interprete them.

Response 4: the Abstract has been revised.

Comments 5: Introduction is Ok. But more references to previous works on the same species or related species regarding to hypoxia responses are needed.

Response 5: Thanks for the suggestion. The Introduction has been revised, and more references have been added.

Comments 6: Methods: Indicate which organ was sampled for analysis.

Response 6: In this study, we took whole body tissue because of the small size of the juvenile snails.

Comments 7: How did the authors measured soluble protein levels in the tissue extract?

Response 7: In this study, the total protein (TP) content were measured using a microplate reader with assay kits purchased from Suzhou Grace Biotechnology Co., Ltd., China, following the manufacturer’s instructions.

Comments 8: Statistical analysis needs revision. The authors had three levels of DO and seven sampling times. Thus they must analyze the results by repeated measure two ANOVA.

Response 8: Thank you for the helpful suggestion. All statistics have been changed to nonparametric two-way analysis using the aligned-rank transform (ART) method. Post-hoc comparisons for interaction decomposition were done using the Holm method. Differences between treatments at each time point and between time points for each treatment group have also been labeled in the figures.

Comments 9: What post hoc test was used for multiple comparison?

Response 9: Post-hoc comparisons for interaction decomposition were done using the Holm method.

Comments 10: Results: Write the exact p-values in the results. Add the name of the statistical test for each figure

Response 10: most p-values have been added in the results. The statistical test methods have been added in the figures.

Comments 11: Discussion: This section needs extensive work. There are some mis-iterpretations. For example, the authors wrote about the lipid peroxidation, but they did not measured any indicators of lipid peroxidation (e.g. MDA and Protein carbonyl). The decrease in SOD activity is likely due to the decrease in metabolic rate, leading to less cell respiration and superoxide ion production.

Response 11: Thanks for the suggestion. We have checked the manuscript and deleted irrelevant content. We agree the reviewer. the Discussion has been revised.

Round 2

Reviewer 1 Report

Comments and Suggestions for Authors

The authors clearly made a good job in improving the data presentation and transparency; yet the data analysis seems still to be an issue at some points. Here are more specific comments:

1) I very much appreciate the updated figures with additional information, as well as more adequate statistical analysis and the raw data table. However, I still suggest providing more details in the text on L208-213:
- What software was used for the analysis?
- The “Holm method” describes the way of correcting p-values for multiple comparisons, but what post-hoc test was used?
- Even more importantly, how the correction for multiple comparisons was applied? In particular, how many pair-wise comparisons were treated together (3 or 21 or some other?) and how many such groupings were used per figure?
- “aligned-rank transform” probably means Aligned Ranks Transformation ANOVA?
- Speaking about figure captions, what does “at p < 0.01 or 0.05” mean? In Methods only p < 0.05 is mentioned.

2) Experimental design and some methods are not described in more details (here I’m mostly citing the Authors’ response):
- “Since the specific time of death of each individual was not recorded, it was not possible to draw survival curves with the Kaplan-Meier method”. It, in fact, is possible to use Kaplan-Meier method precisely in this case, please refer to the first review. But more importantly, the procedure for calculating the mortality rates for the current Fig. 1 is not described in Methods. To my oponion, it must be.
- Authors write: “As stated by the reviewer, because samples were taken every day, the total number of snails decreased each day, and the cumulative mortality rate was calculated assuming that the sampled individuals were still alive”. I do not understand this explanation. Please provide clear description in the manuscript in section 2.2 or maybe 2.6.
- On a related issue, the Authors write on L219-220: “...was significantly higher than...”. The term “significantly” usually implies “statisticaly significantly”, and the Authors use it in this way in the very next paragraph. However, despite the difference on Fig 1 is clear, no statistical analysis was applied to mortality data (it could be with the Kaplan-Meier method). I would recommend more cautios wording here.
- “Response 10: The individuals used for respiration measurements were returned to the same aquaria. Actually, the snails were placed in a mesh bag, so the same set of snails was measured each time”. This information must be in the main text.
- “Response 14: In order to obtain the sequence information of the related genes, we performed a transcriptome analysis. After designing and synthesizing the primers, we then verified the reliability of the primers by PCR and electrophoresis”. This is very vague and not in the manuscript. Could you please elaborate in the section 2.5. If the data are unpublished, a direct link to the deposited data would be fine (in NCBI or analog). Reliability of the primers is a critical issue.

3) The Discussion + Conclusions are still mostly descriptive and do not provide many novel outcomes or direct ecological comparisons with other species. Some important comments on specific points:
- L355 "had consumed all their energy resources to survive". This is not obvious to me; energy resources could still be available, but the animal may just be not able to use the energy. As the most simple possibility, acidic by-products of anaerobic metabolism could shift pH beyond some threshold.
- L379-380 and also L382 "This finding suggests an enhancement in anaerobic metabolism as a means to conserve energy". I do not understand what other options the snail has under hypoxia and why this is a way to conserve energy.
- L452-453 "This finding suggests that hypoxic stress induces oxidative stress in B. areolata". I do not see how a decrease in SOD activity (moreover, explainable by simple decrease in gene expression) indicates oxidative stress. General depression of gene expression under stress is a common phenomenon.
- L468 “findings suggest that B. areolata is sensitive to hypoxic stress”. To my knowledge, the overwhelming majority of animals are sensitive to hypoxic stress. The question is, to what extent in comparison to other species?

Minor issues:
- L20 "immunological functions" and L40 "adequate immunity". As far as I understood, the Authors decided to remove the analysis from immonological point of view from the manuscript.
- L34 “acid phosphatase (ACP)” do you mean AKP here?
- L37 “levels of ACP and AKP genes were significantly downregulated” this is not true for AKP.
- L38 “exhibited elevated SOD gene expression levels under 0.5 mg O 2/L” again, doesn’t match Fig. 5.
- L74-75 "Hyoxia can activate NADPH oxidase,leading to an increase in intracellular reactive oxygen species (ROS) [13]". To my knowledge, this is not the only and not the leading mechanism behind ROS production in hypoxia. Furthermore, [13] is an old experimental paper on cell cultures; a more recent review would be much better suited here.
- L97 “our previous acute hypoxia study” please provide the reference.
- L117 "salinity levels at 28" in what units?
- L178 "The enzyme activity was calculated by measuring the absorbance at 520 nm" It would be good to specify what substance is measured at this wavelength.
- L227 and later, p-values can be presented with ~3 digits after the decimal point or even less sometimes; it's unnecessary to spell all numbers.

Author Response

Response to Reviewer 1 Comments

We greatly appreciate the reviewer's patient, meticulous examination, and constructive suggestions. Please find the detailed responses below and the corresponding revisions in track changes in the re-submitted manuscript (in blue text).

  1. Responses for specific comments

Comments 1: 1) I very much appreciate the updated figures with additional information, as well as more adequate statistical analysis and the raw data table. However, I still suggest providing more details in the text on L208-213:
- What software was used for the analysis?
- The “Holm method” describes the way of correcting p-values for multiple comparisons, but what post-hoc test was used?
- Even more importantly, how the correction for multiple comparisons was applied? In particular, how many pair-wise comparisons were treated together (3 or 21 or some other?) and how many such groupings were used per figure?
- “aligned-rank transform” probably means Aligned Ranks Transformation ANOVA?
- Speaking about figure captions, what does “at p < 0.01 or 0.05” mean? In Methods only p < 0.05 is mentioned.

Response 1: Thank you for the helpful suggestion. We have learned a lot from the comments.

(1) R software (v4.3.0, R Development Core Team 2019) was used for the analysis.

(2) simple effect test was used for post-hoc comparisons.

(3) when there was a significant interaction effect, simple main effects test was conducted to evaluate the effect of one independent variable at specific levels of another independent variable (for time variable at specific levels, n=3; for DO variable at specific levels, n=7).

(4) Yes. “aligned-rank transform” probably means Aligned Ranks Transformation ANOVA. The manuscript has been revised.

(5) Because most p values have been added in the results. We revised the figure captions.

Comments 2: Experimental design and some methods are not described in more details (here I’m mostly citing the Authors’ response):
- “Since the specific time of death of each individual was not recorded, it was not possible to draw survival curves with the Kaplan-Meier method”. It, in fact, is possible to use Kaplan-Meier method precisely in this case, please refer to the first review. But more importantly, the procedure for calculating the mortality rates for the current Fig. 1 is not described in Methods. To my opinion, it must be.
- Authors write: “As stated by the reviewer, because samples were taken every day, the total number of snails decreased each day, and the cumulative mortality rate was calculated assuming that the sampled individuals were still alive”. I do not understand this explanation. Please provide clear description in the manuscript in section 2.2 or maybe 2.6.
- On a related issue, the Authors write on L219-220: “...was significantly higher than...”. The term “significantly” usually implies “statistically significantly”, and the Authors use it in this way in the very next paragraph. However, despite the difference on Fig 1 is clear, no statistical analysis was applied to mortality data (it could be with the Kaplan-Meier method). I would recommend more cautios wording here.
- “Response 10: The individuals used for respiration measurements were returned to the same aquaria. Actually, the snails were placed in a mesh bag, so the same set of snails was measured each time”. This information must be in the main text.
- “Response 14: In order to obtain the sequence information of the related genes, we performed a transcriptome analysis. After designing and synthesizing the primers, we then verified the reliability of the primers by PCR and electrophoresis”. This is very vague and not in the manuscript. Could you please elaborate in the section 2.5. If the data are unpublished, a direct link to the deposited data would be fine (in NCBI or analog). Reliability of the primers is a critical issue.

Response 2: (1) We have drawn Figure 1 using Kaplan-Meier method. Statistical analysis has been performed to mortality data with the Kaplan-Meier analysis.

(2) For the calculation of mortality rate, because 2 snails were taken from each drum and dissected every 24 hours for enzyme activity and gene expression analysis, the total number of snails decreased each day. In calculating the cumulative mortality rate, we included these snails sampled for ease of calculation, assuming that they would live until the end of the experiment. This may underestimate the mortality, but we think that this would also reflect to some extent the effect of different levels of hypoxic stress on the survival of Babylonia areolata.

(3) There are no publicly published sequences or references for most of the target genes analyzed in the experiment. In order to obtain the sequences of the genes, we took the tissue sample of a snail and conducted a De novo assembly transcriptome analysis. After obtaining the Unigene dataset, we obtained the ID information and sequence information of the target genes through annotation information. Primers were designed using Primer Premier 5 software. For each target gene we designed several primers, which were verified by PCR amplification and gel electrophoresis, and primers with high specificity and amplification efficiency were used for real-time PCR analysis. We did not upload these gene sequences to NCBI. The sequence of the target genes has been submitted as a supplementary document. The primer sequence for actin gene were referenced from Feng’s master thesis “Microscopic observation of the tentacles and siphons of Babylonia areolata and study of olfactory genes” (in Chinese) (page 39), which has been cited in the manuscript and submitted as a supplementary document.

Comments 3: The Discussion + Conclusions are still mostly descriptive and do not provide many novel outcomes or direct ecological comparisons with other species. Some important comments on specific points:

Response 3: We are grateful for the suggestion. The Discussion has been revised.
- L355 "had consumed all their energy resources to survive". This is not obvious to me; energy resources could still be available, but the animal may just be not able to use the energy. As the most simple possibility, acidic by-products of anaerobic metabolism could shift pH beyond some threshold.

(1) With regard to the causes of increased mortality, we reviewed the relevant literature and revised this section (Line 365-373).

- L379-380 and also L382 "This finding suggests an enhancement in anaerobic metabolism as a means to conserve energy". I do not understand what other options the snail has under hypoxia and why this is a way to conserve energy.

(2) this section has been revised and the vague expression has been removed (Line 399-400). On the role of anaerobic metabolism in the adaptation of molluscs to low-oxygen environments, many studies have found that the metabolic rates of molluscs decline as dissolved oxygen levels decrease. The decline in metabolic rates is partly due to lower dissolved oxygen concentrations and the inability of shellfish to obtain adequate oxygen. Also, this is a result of shellfish actively reducing their metabolic levels. By reducing metabolic rates through anaerobic metabolism, the rates of all ATP utilizing reactions in cells are strongly suppressed to a level that matches the anoxic rate of ATP production to survive hypoxic stress (Larade and Storey, 2002). Molluscs also employ several metabolic pathways to avoid cellular damage from the accumulation of acidic byproducts.
- L452-453 "This finding suggests that hypoxic stress induces oxidative stress in B. areolata". I do not see how a decrease in SOD activity (moreover, explainable by simple decrease in gene expression) indicates oxidative stress. General depression of gene expression under stress is a common phenomenon.

(3) We agree with the reviewer that decreased enzyme activity is not necessarily associated with oxidative stress. This section has been revised as “These findings indicate a possible inhibition of antioxidative responses in the snails during exposure to low levels of DO, which has been observed in other aquatic animals” (Line 468-470).
- L468 “findings suggest that B. areolata is sensitive to hypoxic stress”. To my knowledge, the overwhelming majority of animals are sensitive to hypoxic stress. The question is, to what extent in comparison to other species?

(4) The Conclusion has been revised, and the sentence has been removed.

  1. Responses for minor comments

Comment 1: L20 "immunological functions" and L40 "adequate immunity". As far as I understood, the Authors decided to remove the analysis from immonological point of view from the manuscript.

Response 1: Thank you for the suggestion. Both have been revised.

Comment 2: L34 “acid phosphatase (ACP)” do you mean AKP here?

Response 2: We are sorry for the mistake. It has been revised.

Comment 3: L37 “levels of ACP and AKP genes were significantly downregulated” this is not true for AKP.

Response 3: Thank you for pointing out our mistake. It has been revised as “levels of ACP and SOD genes were significantly downregulated”.

Comment 4: L38 “exhibited elevated SOD gene expression levels under 0.5 mg O2/L” again, doesn’t match Fig. 5.

Response 4: It has been revised as “exhibited elevated AKP gene expression levels under 0.5 mg O2/L”.

Comment 5: L74-75 "Hyoxia can activate NADPH oxidase, leading to an increase in intracellular reactive oxygen species (ROS) [13]". To my knowledge, this is not the only and not the leading mechanism behind ROS production in hypoxia. Furthermore, [13] is an old experimental paper on cell cultures; a more recent review would be much better suited here.

Response 5: The sentence has been revised, and a review paper has been cited in the manuscript.

Comment 6: L97 “our previous acute hypoxia study” please provide the reference.

Response 6: The study has been published in Chinese, and has been cited in the manuscript. (Alva, R.; Wiebe, J.E.; Stuart, J.A. Revisiting reactive oxygen species production in hypoxia. Pflugers. Arch. - Eur. J. Physiol. 2024, 476, 1423–1444.)

Comment 7: L117 "salinity levels at 28" in what units?

Response 7: the unit of salinity is ‰.
Comment 8: L178 "The enzyme activity was calculated by measuring the absorbance at 520 nm" It would be good to specify what substance is measured at this wavelength.

Response 8: the absorbance of quinone derivatives was measured at 520 nm.

Comment 9: L227 and later, p-values can be presented with ~3 digits after the decimal point or even less sometimes; it's unnecessary to spell all numbers.

Response 9: Thank you for the suggestion. The p-values has been revised and presented with ~3 digits after the decimal point.

Reviewer 2 Report

Comments and Suggestions for Authors

.

Author Response

No comments from reviewer 2